# CLDTLog: System Log Anomaly Detection Method Based on Contrastive Learning and Dual Objective Tasks

**DOI:** 10.3390/s23115042

**Published:** 2023-05-24

**Authors:** Gaoqi Tian, Nurbol Luktarhan, Haojie Wu, Zhaolei Shi

**Affiliations:** 1School of Software, Xinjiang University, Urumqi 830046, China; t_gaoqi@stu.xju.edu.cn (G.T.); 107552104325@stu.xju.edu.cn (H.W.); 2College of Information Science and Engineering, Xinjiang University, Urumqi 830046, China; shizhaolei@stu.xju.edu.cn

**Keywords:** log anomaly detection, contrastive learning, dual objective tasks, bidirectional encoder representation from transformers

## Abstract

System logs are a crucial component of system maintainability, as they record the status of the system and essential events for troubleshooting and maintenance when necessary. Therefore, anomaly detection of system logs is crucial. Recent research has focused on extracting semantic information from unstructured log messages for log anomaly detection tasks. Since BERT models work well in natural language processing, this paper proposes an approach called CLDTLog, which introduces contrastive learning and dual-objective tasks in a BERT pre-trained model and performs anomaly detection on system logs through a fully connected layer. This approach does not require log parsing and thus can avoid the uncertainty caused by log parsing. We trained the CLDTLog model on two log datasets (HDFS and BGL) and achieved F1 scores of 0.9971 and 0.9999 on the HDFS and BGL datasets, respectively, which performed better than all known methods. In addition, when using only 1% of the BGL dataset as training data, CLDTLog still achieves an F1 score of 0.9993, showing excellent generalization performance with a significant reduction of the training cost.

## 1. Introduction

One of the critical features of modern large-scale systems is their need for high availability and reliability [1]. Due to modern systems’ increased size and complexity, anomaly detection plays a crucial role in guaranteeing system reliability and stability. Modern systems usually consist of many sensors, devices, and applications, which may be subject to various interference and attacks during operation, resulting in abnormal system behavior. Anomaly detection technology can effectively monitor and identify these abnormal behaviors, helping system operations and maintenance personnel to find and solve problems promptly [2].

For software-intensive systems, console logs are typically generated to record system status and critical events. These logs provide engineers with detailed information about the system status and help them detect anomalies and determine the cause of failures [3]. However, anomaly detection based on log sequences has become increasingly difficult due to the sheer volume of log data and traditional manual analysis methods’ time-consuming and error-prone nature. With the widespread use of machine learning in many areas of society [4], some log anomaly detection methods [5,6,7] have emerged in recent years. These methods still need more information extraction, and their imprecise analysis affects anomaly detection accuracy.

With the advent of the Big Data era, deep learning has widely penetrated various fields [8]. Due to increased data size and complexity, deep learning is a powerful tool for extracting useful information and patterns from giant datasets [9,10]. Therefore, to overcome the limitations of traditional log anomaly detection methods, researchers [11,12,13,14,15,16] are using deep learning methods extensively. Compared with traditional methods, deep learning can better understand the relationship between different logs in a log sequence, thus enabling more accurate anomaly detection. Although deep learning methods have some problems in practice, it is still a hot topic and development direction for current research.

Existing approaches usually require parsing logs to process semi-structured log data. In this process, the parser removes the variable part from the log information and keeps the constant part for obtaining log events. However, Le et al. [2] showed that log parsing errors can affect subsequent anomaly detection tasks and reduce detection accuracy.

Although existing system log detection methods based on log semantics [13,15] have possessed high robustness and detection effectiveness, they still have some limitations. These methods can only obtain limited log semantic information from the log text and cannot fully extract the deeper semantic information embedded in the logs. In addition, the existing methods need more advanced optimization and processing for the feature information extracted from logs, so there is a need to further improve the characterization capability and accuracy of the methods.

Therefore, this study aims to address these challenges and improve the performance and generalization of anomaly detection. To this end, we propose a method called CLDTLog, which combines contrastive learning and dual-objective tasks, and illustrate the effectiveness of CLDTLog in terms of the following five research questions:

(1) How effective is CLDTLog in anomaly detection? We will evaluate the anomaly detection performance of CLDTLog on different datasets and compare it with existing methods to verify its effectiveness. (2) Does contrastive learning have a significant impact on CLDTLog? We will explore the role of contrastive learning techniques in CLDTLog and analyze its improvement and implications for anomaly detection performance. (3) What is the generalization performance of CLDTLog? We will evaluate the dependence of CLDTLog on labeled data by examining its performance when using training datasets of different sizes and verify its generalization ability in real-world applications. (4) How to choose the appropriate hyperparameters to combine the two target tasks? We will explore how to select the proper hyperparameters to balance between the two target tasks to achieve the best anomaly detection performance. (5) How does the log sequence length affect CLDTLog? We will investigate the impact of log sequence length on CLDTLog performance and analyze its performance on data.

By answering these five key questions, we aim to gain a deeper understanding of the strengths, limitations, and applicability of the CLDTLog approach and provide valuable insights for further research and practice in the area of log anomaly detection.

In CLDTlog, firstly, we group all log messages by timestamps or block_id and then use a pre-trained BERT model that has been fine-tuned by contrastive learning and dual-objective tasks to extract semantic vectors of the entire log sequence. These vectors capture the semantic information embedded in the log sequence and the relationship between each log message. Next, we employ a fully connected layer as a classifier with these semantic vectors as input. Our BERT model is fine-tuned to learn contextual information from log sequences and reduce the similarity between normal and abnormal log sequences, which can be effectively used for log-based anomaly detection tasks.

We evaluate the proposed method using two public datasets by answering the above five questions. The experimental results show that CLDTlog can deeply understand the semantics of log data. The method achieves high F1 scores (both close to 1) for anomaly detection, outperforming existing log anomaly detection methods. Our contributions:(1)We propose CLDTLog, which treats log sequences as natural language sequences, fine-tunes the BERT model [17] using the idea of contrastive learning and dual-objective tasks to obtain semantic representations of log sequences, and then uses a fully connected neural network in order to detect anomalies.(2)The contrastive learning we use increases the semantic distance between the semantic features of normal and abnormal log sequences, which makes it easier for the model to distinguish abnormal log sequences from regular log sequences, thus improving the accuracy of abnormality detection.(3)Our proposed method achieves an F1 score of 0.9993 on the BGL dataset using only 1% of the dataset for training, showing powerful generalization performance while reducing the training cost in practical applications.

## 2. Related Work

### 2.1. Log Parsing

The accuracy of log parsing has a relatively significant impact on the performance of log mining [18]. Log parsers may produce inconsistent results depending on the preprocessing steps and parameter sets [19]. Therefore, optimizing the preprocessing step can further improve the accuracy of log parsing [18], although this requires some additional manual work [19], which is also very important. Nedelkoski et al. [20] enhanced the evaluation protocol by adding a new metric to demonstrate the offset between the generated template and the actual log message type. Their proposed Nulog and 12 other log parsers were compared on ten datasets. The experimental results show that NuLog outperforms other log parsers. Other better-performing log parsers include Drain [21], IPLoM [22], AEL [23], and Spell [24]. However, some of these parsers have model parameters that need to be manually tuned, and similarly, some models need to adapt better to the log volume. Therefore, choosing the appropriate log parser and the corresponding parameters for different datasets and application scenarios is important [2].

In recent years, with the emergence of new log types [16], new log events often appear due to the evolution of the software, making the log data unstable. Therefore, Zhang et al. [15] pointed out the need to use some strategies to deal with these unstable log data. Their empirical study of the Microsoft Online Services system showed that up to 30.3% of the logs were changed in the latest version. Thus, handling unstable log data is an important challenge for log mining tasks. In this regard, smarter log preprocessing methods must be developed to improve the accuracy and stability of log parsing.

### 2.2. Log Representation and Anomaly Detection

In recent years, most log-based anomaly detection methods have used log parsers to represent log messages as log events. However, deep learning-based models have become a hot research topic. DeepLog [11], proposed by Du et al., learns the normal execution of the system by predicting the following log event and detects anomalies by comparing the incoming log events with the prediction results of the LSTM (Long Short Term Memory) model. However, the method cannot handle unknown log events. To solve this problem, Zhang et al. [15] combined pre-trained FastText [25] with TF-IDF weights to learn the representation vector of log templates and then they used an attention-based Bi-LSTM (Bi-directional Long Short-Term Memory) model to detect anomalies. Experimental results show that LogRobust can solve the problem of instability of log events and the approach ensures that appropriate log template updates do not cause significant changes to the generated semantic features [26].

LogAnomaly [16] and SwissLog [27] are two different methods for processing log messages. LogAnomaly uses FT-Tree [28] to parse log messages into templates and then encodes them using template2Vec based on Word2Vec [29]. SwissLog, on the other hand, applies a dictionary-based approach to parse log messages and obtain their semantic information.

Due to the limitations of existing methods, the semantics of log messages may not be captured and produce incorrect results. Log2Vec [30] is a method to convert raw log messages into semantic vectors, but it cannot effectively handle some domain-specific words [31] because it utilizes character-level features. In addition, the approach uses a word2vec-based model that ignores the contextual information in the sentences [2,27] and cannot fully understand the semantics of log messages.

To address these limitations, we propose an anomaly detection method called CLDTLog. It fine-tunes the BERT model using contrastive learning and dual-objective tasks to understand the semantics of raw log messages and capture contextual information without using a log parser. Combined with a fully connected neural network, CLDTLog achieves optimal anomaly detection accuracy. In addition, we only use the content part of the log messages from the target system without any auxiliary data, which can further improve the utility.

## 3. Background

### 3.1. BERT

The full name of BERT is Bidirectional Encoder Representations from Transformers, which uses a bidirectional encoder based on the Transformer architecture to learn language knowledge through a large-scale unsupervised pre-training task, which is then fine-tuned by a supervised task to adapt to specific downstream natural language processing tasks. The pre-training tasks for BERT models typically include two approaches: Masked Language Model (MLM) and Next Sentence Prediction (NSP). In the MLM task, the BERT model masks some words in the input text and then predicts what those masked words are. In the NSP task, the BERT model determines whether two input texts are adjacent sentences and predicts whether they are semantically related. Both tasks help the BERT model to learn a more comprehensive and rich linguistic representation.

The BERT model has several advantages:(1)The BERT model employs a bidirectional encoder, which can effectively capture contextual information in the input text.(2)The BERT model learns language knowledge through large-scale unsupervised learning, which can reduce the reliance on large amounts of labeled data.(3)The BERT model is highly scalable and can be applied to various natural language processing tasks, such as text classification, named entity recognition, language generation, etc.(4)The BERT model has achieved state-of-the-art performance on several natural language processing tasks, proving its solid practical value in natural language processing.

Therefore, we use BERT as our feature extractor.

### 3.2. Focal Loss

Focal Loss [32] is a loss function designed for the category imbalance problem. Compared with the traditional cross-entropy loss, Focal Loss improves the recognition ability of the model for hard-to-classify samples by introducing adjustment factors to reduce the loss weight for easy-to-classify samples and enhance the weight of hard-to-classify samples [33]. This loss function has achieved significant optimization results in many practical tasks.

Specifically, Focal loss introduces a modifier γ to adjust the weights of the samples so that the hard-to-classify samples have larger weights and the easy-to-classify samples have smaller weights. The Focal loss is shown in Formula (Equation 1).
(1)$$L_{FL}=-{\left(1-p_{t}\right)}^{\gamma }log\left(p_{t}\right)$$

Among them, $p_{t}$ denotes the confidence level calculated from the model prediction and the actual label, which can be expressed by Formula (Equation 2).
(2)$$p_{t}=\left\{\begin{array}{cc} p & \;\mathrm{if}\;y\;=\;1\\ 1-p & \;\mathrm{otherwise} \end{array}\right.$$
where *p* is the probability of the model prediction, and y represents the actual label of the input sample, where 1 indicates a positive class, and 0 indicates a negative class. When γ is 1, the Focal Loss degenerates to the standard cross-entropy loss function.

In text classification, Focal loss can reduce the loss of common categories that are easy to classify, thus improving the classification accuracy of rare categories or complex samples. This is particularly effective for dealing with text data with long-tail distribution [34]. Focal loss can also deal with category imbalance in multi-label text classification tasks. Adjusting the attenuation factor can balance the weights of different categories and thus improve the accuracy for rare categories or complex samples [35].

In our method, the different number of samples in different categories in the datasets may lead to poor recognition of the model for the classes with fewer samples, so we use Focal loss as the first objective task to solve the problem of category imbalance in the datasets.

### 3.3. Triplet Loss

The Triplet loss [36] is a widely used loss function in face recognition tasks. The purpose is to do the differentiation of similar samples from different categories. Therefore, the advantage of Triplet loss is detail differentiation, i.e., when two inputs are similar, Triplet loss can better model the details, which is equivalent to adding a measure of the difference between the two inputs and learning a better representation of the inputs [37].

Specifically, the Triplet loss is defined by constructing triplet data; each triplet data consists of an anchor sample, a positive sample, and a negative sample. The anchor sample and positive sample belong to the same category, while the negative sample belongs to a different category from the anchor sample. Triplet loss tries to make the distance between the anchor sample and the positive sample as small as possible while driving the distance between the anchor sample and the negative sample as large as possible. In addition, to keep the features of the samples from aggregating into a tiny space, it is required that the negative and positive samples should be at least farther apart than the two positive samples.

Therefore, it is hoped that Formulas (Equation 3) and (Equation 4) will hold:(3)$$||f\left(x_{i}^{a}\right)-f\left(x_{i}^{p}\right){\left|\right|}_{2}^{2}+\alpha <||f\left(x_{i}^{a}\right)-f\left(x_{i}^{n}\right){\left|\right|}_{2}^{2}$$
(4)$$\forall (f\left(x_{i}^{a}\right),f\left(x_{i}^{p}\right),f\left(x_{i}^{n}\right))\in T$$

Among them, $f(x_{i}^{a})$ is the feature representation of the anchor sample, $f(x_{i}^{p})$ is the feature representation of the positive sample, which belongs to the same class as the anchor sample, $f(x_{i}^{n})$ is the feature representation of the negative sample, which belongs to a different class from the anchor sample, α is the boundary threshold (margin), which is used to control the distance between positive and negative samples, *T* represents the entire feature space. Then, according to Formulas (Equation 3) and (Equation 4), Triplet Loss is shown in Formula (Equation 5).
(5)$$L_{TL}=\sum_{i}^{N}(||f\left(x_{i}^{a}\right)-f\left(x_{i}^{p}\right){\left|\right|}_{2}^{2}-||f\left(x_{i}^{a}\right)-f\left(x_{i}^{n}\right){\left|\right|}_{2}^{2}{+\alpha )}_{+}$$
where $L_{TL}$ is the overall loss function of Triplet loss, *N* is the number of sample pairs, and the symbol “+” in the lower right corner of the above equation indicates that the loss is an actual loss when greater than 0; otherwise, the loss is 0.

In face recognition, Triplet loss is used to learn discriminative feature representations of faces. Triplet loss can drive models to learn feature spaces that make the same person’s image features closer and those of different people more dispersed. Such feature representation can improve the accuracy and robustness of face recognition systems [38]. Triplet loss is also widely used to learn matching models in text-matching tasks, such as question-and-answer systems or information retrieval. The performance of tasks such as text similarity computation, sentence matching, and information retrieval can be improved by learning to map matching text pairs to regions in the feature space that are closer together and mapping mismatched text pairs to regions that are farther away [39].

Contrastive learning is a deep learning idea, the core idea of which is to learn to make feature representations more similar between samples of the same category and more different between samples of the different categories. To achieve this goal, we use the Triplet loss to implement contrastive learning as the second objective task to learn high-quality feature representations.

## 4. The Proposed Methodology

### 4.1. Log Preprocessing

Building the model requires first preprocessing the logs into a form of data that can be fed into the model. Therefore, log preprocessing is the first step. For both BGL (BlueGene/L) and HDFS (HDFS distributed file system) datasets, we first partition the original log datasets we collected to obtain the content part of the log messages. Then, the log message content is processed. First, the log message is split into words using the separator as a marker. Then, to avoid the effect of upper and lower case letters on the log message, each upper case letter is converted to lower case. All special characters that are not partitioned characters are removed from the log statement content. These non-characters include operators, punctuation marks, and numbers. This type of token is removed because it usually indicates a variable in the log message and has no information. Figure 1 shows the process of BGL log preprocessing.

For the BGL log dataset, log sequences are obtained by grouping logs according to different window sizes after processing. Figure 2 demonstrates two log sequences consisting of 7 log contents after a window size of 5 and a sliding window of 2. If one message of the log sequences is an abnormal log, then this log sequence is treated as an abnormal log sequence. The label of the HDFS log dataset is determined based on whether the log sequences formed by grouping block_id are abnormal or not. Therefore, the HDFS log dataset is organized by block_id to form log sequences.

### 4.2. Log Sequence Representation

In the log preprocessing phase, we obtain the log sequence, and in the log sequence representation phase, we need to tokenize the sequence using BERT’s word splitter. BERT uses a WordPiece tokenizer [40], which can split words into subwords or characters. The WordPiece tokenizer has the advantage that it can handle unknown words to obtain more accurate semantic information and can reduce the size of the vocabulary. For example, “NameSystemaddStoredBlock” is split into [‘Name’, ‘##System’, ‘##add’, ‘##Stored’, ‘##Block’], where the ‘##’ prefix indicates that the subword is part of a word and not a separate word. Each log sequence is converted into a set of words and subwords. For each subword, we convert it to the corresponding ID according to the vocabulary and then add special tokens, such as [CLS] and [SEP], at the beginning and end of the input sequence to mark the beginning and end of the sentence. Finally, we unify the length of the input sequence to a fixed length by filling it by adding [PAD] tokens at the end of the sequence or truncating the log sequence.

Next, as shown in Figure 3, we take the processed log sequences through token embedding, segment embedding, and positional embedding to obtain the semantic vectors of the log sequences, which are encoded in the log encoding layer. The log encoder uses a transformer-based bidirectional encoding structure consisting of 12 encoders. Each encoder consists of a multi-head attention layer, residual block layer, norm layer, and feedforward layer [41].

### 4.3. CLDTLog

The BERT model has been trained with a large-scale corpus to learn the general features of the language. To further reduce the training time, we only tune the last three encoders of the BERT pre-trained model. This approach not only improves training efficiency but also achieves excellent results. There are two target tasks in our approach.

The purpose of the first task is to reduce the similarity between the semantic features of normal and abnormal log sequences so that the semantic features generated by normal and abnormal log sequences are farther apart in the semantic space [26]. Completing the first task requires us to construct the Triplet loss by obtaining vector representations of anchor samples, positive samples, and negative samples in the output layer of BERT. In a batch, all samples can be used as anchor samples. When selecting an anchor sample, the sample different from the anchor but in the same category can be used as the positive sample. The sample differs from the anchor sample and can be used as the negative sample. The first target task can help CLDTLog improve anomaly detection accuracy.

The second task is to make the actual prediction result close to the desired output. Here, we add a fully connected layer as a classifier after the BERT model, and we use Focal loss as the loss function of the classifier to improve the classification accuracy. The second task is used to reduce the weight of easy-to-classify samples and enhance the importance of hard-to-classify samples by introducing a moderator. This improves the recognition ability of the model for hard-to-classify samples. This loss function has been used in many practical tasks to achieve significant optimization results.

System logs are a crucial component of system maintainability. In CLDTLog, the final loss function is constructed by calculating the weighted sum of Triplet loss and Focal loss to improve the performance and reliability of detection. Specifically, their advantages include the following:(1)Improving the robustness of the model: The first task of CLDTLog helps the model to learn the difference between normal and anomaly log sequences better, thus improving the accuracy and recall of detection.(2)Solving the sample imbalance problem: In system log anomaly detection, the number of normal logs is often much larger than the number of anomaly logs, which leads to the sample imbalance problem. The second objective task of CLDTLog can balance the number of normal and abnormal samples by assigning higher weights to anomalous samples. Likewise, it can alleviate the sample imbalance problem by reducing the consequences of easy-to-classify samples, so the model focuses more on classifying complex samples. Thus, the performance of anomaly detection is improved.(3)Adaptability: CLDTLog can be applied to different types of models and tasks with high adaptability.

In summary, CLDTLog can solve the sample imbalance problem, improve the robustness and reliability of the model, and is highly adaptable.

The training and testing phases of CLDTlog are shown in Figure 4. In the training phase, during the log preprocessing, we first process the log data into log sequences as the input data of our model; during the fine-tuning of the BERT model, for the first task, it is a challenging task to choose the adequate positive sample and negative sample. In finding samples, we find similar samples among samples of the same category but not remarkably similar. We also find different samples among samples of different categories but somewhat similar so that our loss will not be 0. We can also achieve the goal of fast convergence. Then, we select three samples (log sequences) and input them to BERT to obtain log sequence representations to calculate the Triplet loss. The second task is to obtain the classification loss. We only use the anchor samples (log sequences) to input into the same BERT model (sharing parameters with BERT of the first task). Then, we add a linear layer to obtain the classification loss after obtaining the log sequence representation. Since most system logs are normal logs, our dataset is unbalanced, so we use Focal loss to calculate the classification loss. Finally, we use the Triplet loss and the classification loss (Focal loss) weighted sum as the final loss function of CLDTLog to tune the model.

We only use the second task in the testing phase to obtain the final classification results. First, we process the logs into log sequences, then input the log sequences into the fine-tuned BERT model, and input the log sequence representations obtained from the BERT model into the linear classifier to obtain the final prediction results.

## 5. Evaluation

### 5.1. Experimental Settings

#### 5.1.1. Research Questions

In this section, we answer the five research questions posed in the introduction to fully evaluate and analyze the performance and effectiveness of our approach in anomaly detection tasks. These research questions include:

**RQ 1**: How effective is CLDTLog in anomaly detection?

**RQ 2**: Does contrast learning have a significant impact on CLDTLog?

**RQ 3**: What is the generalization performance of CLDTLog?

**RQ 4**: How to choose the appropriate hyperparameters to combine the two target tasks?

**RQ 5**: How does the log sequence length affect the CLDTLog?

We will conduct experiments in the order of these research questions and provide detailed experimental designs, method descriptions, and analysis of results to answer these questions and demonstrate the performance and advantages of our method.

#### 5.1.2. Datasets

In this paper, we evaluate the performance of CLDTLog on two public datasets, HDFS and BGL (Blue Gene/L). The HDFS (Hadoop Distributed File System) [42] dataset generated by over 200 Amazon EC2 nodes contains 11,175,629 log messages, which form different log windows (log blocks) based on their block_id, reflecting the execution of a program in the HDFS system. This dataset has 16,838 log windows (2.9%) that indicate system anomalies. In contrast to HDFS, the BGL (Blue Gene/L) [43] dataset, a supercomputing system log dataset collected by Lawrence Livermore National Laboratory (LLNL), contains 4,747,963 log messages [44], of which 348,460 log messages (7.3%) are manually marked as anomalous, while others are marked as normal. Table 1 shows statistics of the data used in the experiments.

#### 5.1.3. Evaluation Indicators

To evaluate the effectiveness of CLDTLog in anomaly detection, we use precision, recall, and F1-score metrics.

TP (True Positives): True cases, where the prediction is positive, and the case is positive.

FP (False Positives): False Positives, predicted to be positive but negative.

FN (False Negatives): False negative cases, predicted to be negative but positive.

Precision: The percentage of correctly detected abnormal log sequences by pattern to all detected anomaly log sequences. The Precision is calculated as shown in Formula (Equation 6).
(6)$$Precision=\frac{TP}{TP+FP}$$

Recall: The percentage of log sequences correctly identified as anomalous to all true anomalies. The recall is calculated as shown in Formula (Equation 7).
(7)$$Recall=\frac{TP}{TP+FN}$$

F1-score: The summed average of precision and recall. The F1-score is calculated as shown in Formula (Equation 8).
(8)$$F1-score=\frac{2*Precision*Recall}{Precision+Recall}$$

### 5.2. RQ1: How Effective Is CLDTLog in Anomaly Detection?

Our research question 1 is to evaluate the effectiveness of CTDTLog on two public log datasets. After preprocessing the raw logs, we found that most of the log messages in both datasets are less than ten characters. For the BGL dataset, the data are labeled according to the log messages. Therefore, we use a single log message for anomaly detection. So, we set the maximum length of the input sequence to 10. To construct log sequences on the HDFS dataset, we associate log messages with the same block_id. Since the dataset is labeled according to the block_id, and most of the log sequences have less than 30 log messages, we set the input sequence length to 300 to ensure that the model can cover most of the log message content.

After that, we separate normal and abnormal log sequences and then extract 80% of the log data from each log sequence as the training set. We divide the log data according to the timestamps of the logs, using the first 80% of the log sequences as the training set and the remaining 20% as the test set. This design ensures that no data leakage occurs.

We compared the results of our proposed approach with seven existing approaches, traditional approaches such as SVM [7], which convert log sequences into log count vectors, and then three different classifiers were used to model the anomaly detection: Approaches that exploit the relationships between logs in a log sequence, such as DeepLog [11], which use an LSTM model to learn sequence relationship between normal logs, which detect anomalies by determining whether to violate the normal log sequence relationship; approaches that exploit semantics such as LogRobust [15] integrate a pre-trained Word2vec model to learn log semantic information, and then, LogRobust uses an attention-based Bi-LSTM to understand and detect anomalies.

Table 2 shows the performance comparison of our seven benchmark models and CLDTLog on the HDFS dataset. From the table, it can be seen that the F1 scores of all methods are 0.94 and above. This is because HDFS is a more stable dataset, and the test set contains a few unknown log templates, so all the proposed methods can obtain high performance on the HDFS log dataset.

However, we can also clearly see the superiority of CLDTLog, as seen from Table 2, that the F1 score distribution of other methods is in the range of 0.94–0.98. At the same time, CLDTLog reaches the optimum in all three evaluation metrics, and they are all extremely close to 1. This illustrates the effectiveness of our contrastive-based learning and dual-objective approach. It also shows that the abnormal log sequences CLDTLog detects are genuine and rarely missed.

The experimental results on the BGL dataset in Table 3 conclude that the Loganomaly, Deeplog, and SVM methods, which perform well on the HDFS dataset, have F1 scores that decrease by 59.6%, 54.3%, and 52.1%, respectively, on the BGL dataset. This is because the BGL dataset is unstable. The SVM method only considers the log count vector, which fails to process unknown log templates and cannot obtain log information, resulting in poor performance. Similarly, Deeplog relies solely on the temporal information of log sequences and cannot achieve good results on the BGL dataset. Although Loganomaly adds positive and negative words based on Deeplog, it still performs poorly on the BGL dataset. Other methods that utilize the semantic information of log sequences can handle unknown log events well and are robust, thus achieving higher performance on the unstable BGL log dataset. Among them, Neurallog also utilizes a pre-trained BERT model as a semantic feature extractor for log sequences and then uses a Transformer-based classifier to classify them, resulting in an F1 score of 0.98. However, it does not adjust the parameters of the BERT model, and the Transformer-based classifier is computationally expensive. Our proposed method, CLDTLog, only adds a linear classifier to the BERT model and trains the model using contrastive learning and a dual-objective task, achieving an F1 score of 0.9999 (≈1) on the BGL dataset. This demonstrates the superiority of CLDTLog over Neurallog and indicates the effectiveness of the proposed approach.

After the experiments, we found that the F1 score of CLDTLog is exceptionally close to 1 on the BGL dataset, which is 0.0028 higher than that on the HDFS dataset. The main reason is that CLDTLog pays more attention to the semantic information of log messages. In the HDFS dataset, most of the log sequences have more than 20 log messages, and if one log message in the log sequence is an anomaly, this log sequence is considered an anomaly. In contrast, one log message in BGL is a log sequence. Therefore, on the BGL dataset, CLDTLog can detect anomalies more accurately compared to the HDFS dataset. This is why CLDTLog can improve the F1 score to 0.9999 on the BGL dataset. In addition, the experimental results in RQ5 also verify this description.

### 5.3. RQ2: Does Contrastive Learning Have a Significant Impact on CLDTLog?

To evaluate the impact of comparative learning on the performance of CLDTLog, we choose the log sequence partitioning method and dataset segmentation method for HDFS and BGL datasets in RQ1 to evaluate CLDTLog and CLDTLog_uncl.

CLDTlog_uncl: Without contrastive learning, the loss function only uses Focal loss.CLDTLog: Contrastive learning is implemented through the Triplet loss function, using the weighted sum of Focal loss and Triplet loss as the loss function.

As seen from Figure 5 and Figure 6, a performance gap exists between CLDTLog_uncl and CLDTLog on the BGL and HDFS datasets, where the CLDTLog method performs significantly better than the CLDTLog_uncl method. This indicates that our proposed method (based on contrastive learning and dual-objective tasks) outperforms the single-task no-contrastive learning method on both datasets in all three metrics. Likewise, it shows that contrastive learning enables the model to reduce the semantic distance between features of similar samples in the feature space and increase the semantic distance between features of different classes of samples in the feature space, making it easier for the model to identify abnormal log sequences.

### 5.4. RQ 3: What Is the Generalization Performance of CLDTLog?

In RQ3, we evaluated the models CLDTLog and CLDTog_uncl on the BGL dataset, using different proportions of training data for evaluation. The statistics of the BGL dataset are shown in Table 4.

In the BGL dataset, except for the training set, the remaining data were used as the test set for model evaluation. As shown in the figure, the solid line indicates the performance of CLDTLog on the three evaluation metrics, and the dashed line indicates the performance of CLDTLog_uncl on the three evaluation metrics. From Figure 7, it can be seen that the performance scores of both models on the three metrics trend upward and have reached high scores as the proportion of training data selected in the BGL dataset increases. However, the two models perform similarly in accuracy, while there is a significant gap in the other two metrics, indicating that comparative learning is effective. It is worth noting that when CLDTLog uses only 1% of the BGL dataset (91.3% of the unknown log templates) as the training set, the F1 score reaches 0.9993, indicating that our model has good generalization performance and can discriminate novel log templates well. In the current situation of high labeling costs, our proposed model has good generalization performance, significantly reducing the cost of training models and having great practical significance.

### 5.5. RQ 4: How to Choose the Appropriate Hyperparameters to Combine the Two Target Tasks?

In this RQ, our goal is to investigate how to combine two target tasks to maximize the performance of CLDTLog. The approach we adopt involves a loss function. As shown in Formula (Equation 9).
(9)$$L=\alpha *L_{TL}+\left(1-\alpha \right)*L_{FL}$$

The $L_{TL}$ in the above equation is the Triplet loss, and $L_{FL}$ is the Focal loss.

We used the grid search method to explore the optimal weight α on the BGL dataset. Because CLDTLog reached a high F1 score when using 1% of the BGL dataset as the training set, we used the top 1% of the BGL dataset as the training set and the rest as the test set for the experiments. When α equals 0 represents using only Focal loss, degenerating to CLDTLog_uncl method, α equals 1 means using only Triplet loss for the experiment. According to the experimental results, CLDTLog can hardly learn anything when the α is 1 and the F1 score is only 0.0565. Therefore, this data point can be disregarded when plotting the figure. We analyze the effect of the remaining α values on CLDTLog performance by plotting Figure 8. The figure shows the precision, recall, and F1 score variation with the α value.

The results show that the precision of CLDTLog gradually increases as the α value increases and reaches its highest point. The recall peaks at an α value of 0.2 and fluctuates slightly later. The F1 score, which combines precision and recall, shows a similar trend.

The combined analysis shows that CLDTLog performs best at an alpha value of 0.2, with high precision, recall, and F1 score. Therefore, we recommend choosing this α value for practical applications to obtain the best performance results.

### 5.6. RQ 5: How Does Log Sequence Length Affect CLDTLog?

In the fifth RQ, we only experiment with the BGL dataset. We set the input sequence length of the BERT model (log sequence length of 1) to 10. We truncate the log sequence length when it is larger than the input sequence length, and we fill it when the log sequence length is smaller than the input sequence length. When we change the log sequence length, we increase the input sequence length of the BERT model accordingly so that the model can cover the same size of data. This is done to exclude the effect of truncating too much log content since the input sequence of the BERT pre-trained model is 512 at maximum. Therefore, when the log sequence length is 100, the input sequence can only be set to 512. Table 5 shows the statistics of input sequence length and log sequence length.

It is clear from Figure 9 that the performance of CLDTLog is negatively correlated with the log sequence length. Moreover, when the sequence length is less than or equal to 50, the performance difference of CLDTLog is not significant. However, the performance degrades more at the sequence length of 100, which is influenced by the truncation of too much valid information limited by the maximum length size of the input sequence.

In this research problem, by comparing the performance of CLDTLog with different log sequence lengths, it is found that CLDTLog performs best when the log sequence length is 1. Therefore, CLDTLog with the log sequence length of 1 was selected as the optimal model for evaluation in the above research problem on the BGL dataset.

## 6. Conclusions

This paper proposes a CLDTLog method based on contrastive learning and dual-objective tasks for automatically detecting log anomalies. Compared with previous LSTM and CNN-based methods, CLDTLog can better capture semantic information from raw logs. In addition, by learning semantic relations extracted from log sequences comparatively, CLDTLog increases the difference between normal and abnormal log sequences compared to the BERT pre-trained-based approach, while using a Focal loss objective function to focus optimization targets on hard-to-classify samples, allowing the model to detect abnormalities in log sequences more easily. We evaluate our proposed method on two log datasets. The experimental results show that CLDTLog outperforms existing Syslog-based anomaly detection methods and has strong generalization performance to reduce the cost of data collection and processing. As a result, when faced with increasingly large log files, our CLDTLog can quickly and accurately filter out anomalous logs, identify potential problems or security threats, and take necessary measures promptly. It also avoids the workload of manually checking line by line.

In future research, we will explore using unsupervised methods for anomaly detection in system logs. Although our approach CLDTLog can achieve efficient anomaly detection performance when using limited samples, it still relies on labeled training data. In practical applications, we may face data labeling limitations or cost challenges. Therefore, we will develop more autonomous and unsupervised techniques to reduce the dependence on labeled data and improve the pervasiveness and scalability of anomaly detection. This will help better adapt to various environments and application scenarios and address data labeling issues that may be encountered in practice.

## Figures and Tables

**Figure 1 sensors-23-05042-f001:**
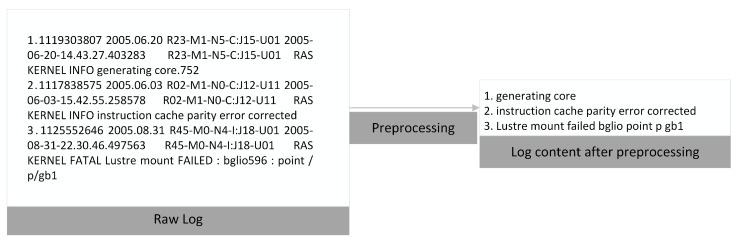
Preprocessing of log messages in the BGL dataset.

**Figure 2 sensors-23-05042-f002:**
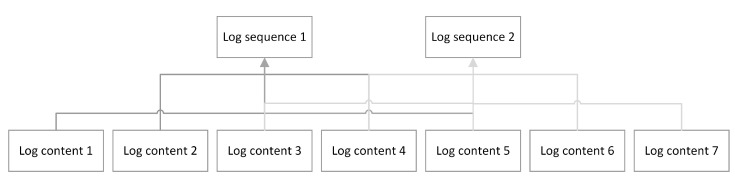
Process the contents of log messages in the BGL dataset into log sequences.

**Figure 3 sensors-23-05042-f003:**
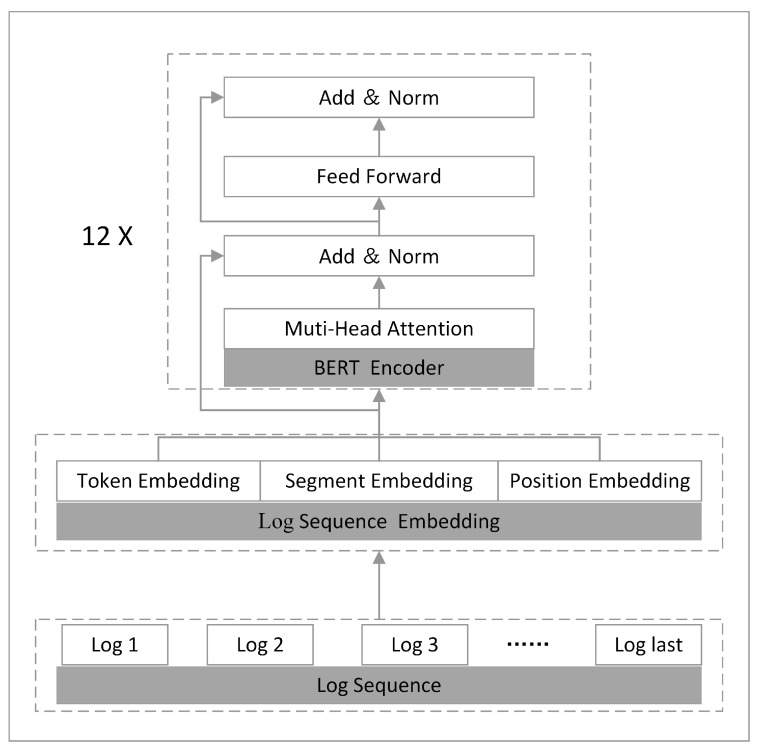
Obtaining log sequence representations.

**Figure 4 sensors-23-05042-f004:**
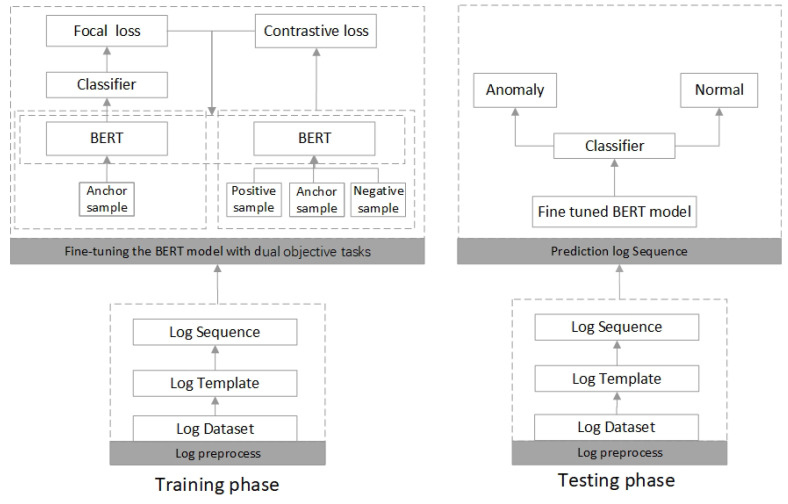
Training and testing process of CLDTLog.

**Figure 5 sensors-23-05042-f005:**
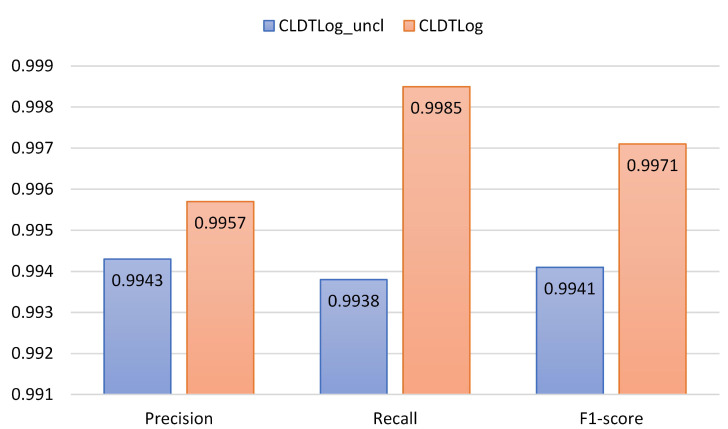
Performance comparison of CLDTLog and CLDTLog_uncl on HDFS dataset.

**Figure 6 sensors-23-05042-f006:**
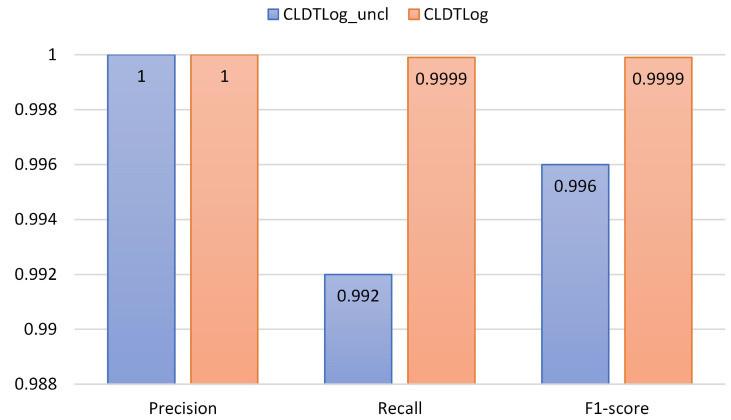
Performance comparison of CLDTLog and CLDTLog_uncl on BGL dataset.

**Figure 7 sensors-23-05042-f007:**
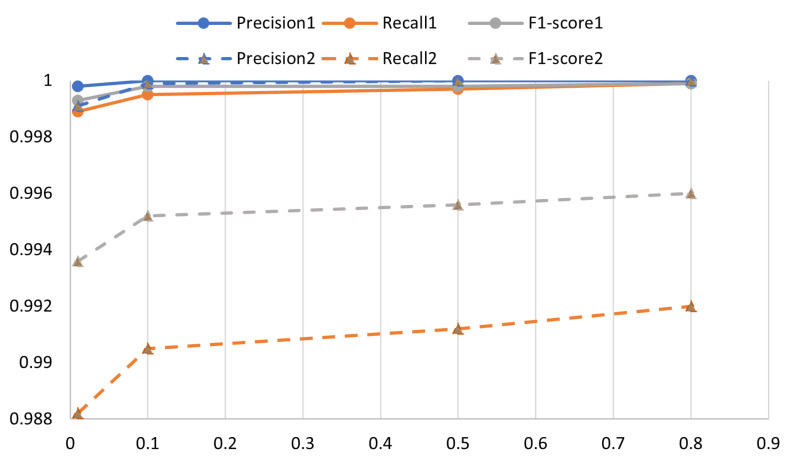
Performance of CLDTLog and CLDTLog_uncl on the BGL dataset with different training ratios selected.

**Figure 8 sensors-23-05042-f008:**
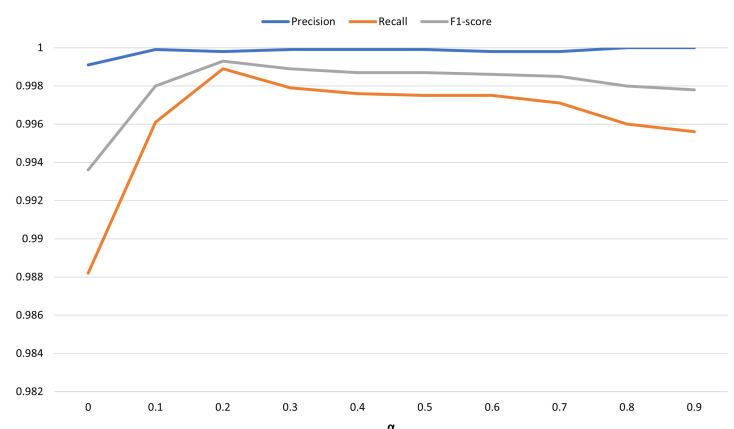
Exploring the optimal hyperparameter α on the BGL dataset.

**Figure 9 sensors-23-05042-f009:**
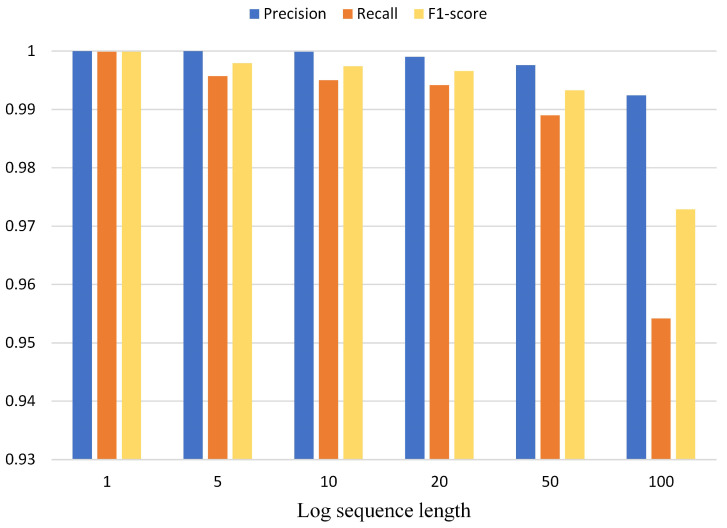
Effect of different log sequence lengths on CLDTLog in the BGL dataset.

**Table 1 sensors-23-05042-t001:** Statistics of the data used in the experiments.

Dataset	Log Event	Grouping	Log Sequence	# Anomaly
HDFS	48	Session	565,061	16,838 (2.9%)
BGL	601	1 log	4,723,906	348,460 (7.3%)
		5 logs	948,782	77,480 (8.2%)
		10 logs	472,391	39,760 (8.4%)
		20 logs	236,196	20,633 (8.7%)
		50 logs	94,879	8975 (9.5%)
		100 logs	47,240	4861 (10.3%)

**Table 2 sensors-23-05042-t002:** Performance comparison of different methods on HDFS dataset.

Method	Precision	Recall	F1 Score
SVM	0.99	0.94	0.96
LogAnomaly	0.90	0.99	0.94
Deeplog	0.88	1.00	0.94
HitAnomaly	1.00	0.97	0.98
CNN	0.97	1.00	0.98
NeuralLog	0.96	1.00	0.98
LogRobust	0.93	1.00	0.96
CLDTLog	0.9957≈1	0.9985≈1	0.9971≈1

**Table 3 sensors-23-05042-t003:** Performance comparison of different methods on BGL dataset.

Method	Precision	Recall	F1 Score
SVM	0.97	0.30	0.46
LogAnomaly	0.31	0.80	0.48
Deeplog	0.27	0.99	0.43
HitAnomaly	0.95	0.90	0.92
CNN	0.87	0.95	0.91
NeuralLog	0.98	0.98	0.98
LogRobust	0.99	0.94	0.97
CLDTLog	**1.0000**	0.9999≈1	0.9999≈1

**Table 4 sensors-23-05042-t004:** Statistics of the BGL dataset.

Train Ratio	Number of Logs	Number of Log Templates
1%	47,239	52
10%	472,390	144
50%	2,361,953	261
80%	3,779,124	414
100%	4,723,906	601

**Table 5 sensors-23-05042-t005:** Statistics of input sequence length and log sequence length.

Log Sequence Length	Input Sequence Length
1	10
5	50
10	100
20	200
50	500
100	512

## Data Availability

The data that support the findings of this study are openly available in loghub at https://doi.org/10.5281/zenodo.3227177.

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
