# Peer review of "CLDTLog: System Log Anomaly Detection Method Based on Contrastive Learning and Dual Objective Tasks"

_sensors, 2023, doi:10.3390/s23115042_

Round 1

Reviewer 1 Report

This paper presents a System Log Anomaly Detection Method. The key contribution claimed in the manuscript is the BERT fine-tuned abnormality detection model using the idea of contrast learning via a triplet-loss and a dual-objective task (focal loss). In addition, the method achieved satisfactory performance.

This paper is generally well-written and structured. The proposed approach is evaluated and presented by answering five research questions. 

However, please thoroughly check the English expression, grammar, and formatting in the manuscript before officially accepting, for example:

  1. The sentence in lines 120-123 of section 2.2 is too long to understand.
  2. Please check the Capitalization Rule of vocabulary in the manuscript, for example. ‘Second’, in line 189.
  3. Please specify all the meanings of tokens in all formulas.

Reviewer 2 Report

This manuscript proposes a Machine learning (ML) method for automatically detecting log anomalies based on contrast learning. I believe the manuscript is suitable for publication in the journal after some minor adjustments.” The authors need to add more justification in the background for the motivation to add a new learning method for the log anomaly detection problem. They need to add also some different ML applications discussion with their difference in a building structure to enhance the literature review, which in my opinion, is very weak, as:

Shinde, Pramila P., and Seema Shah. "A review of machine learning and deep learning applications." 2018 Fourth international conference on computing communication control and automation (ICCUBEA). IEEE, 2018

Moussa, Ghada S., and Mahmoud Owais. "Pre-trained deep learning for hot-mix asphalt dynamic modulus prediction with laboratory effort reduction." Construction and Building Materials 265 (2020): 120239.

Qiao, Lei, et al. "Deep learning based software defect prediction." Neurocomputing 385 (2020): 100-110.

Moussa, Ghada S., Mahmoud Owais, and Essam Dabbour. "Variance-based global sensitivity analysis for rear-end crash investigation using deep learning." Accident Analysis & Prevention 165 (2022): 106514.

In the experimental section: 1. The research question should be moved to be discussed in the introduction section.

2. I think that the improvement in the accuracy is a slight value, the authors need to justify the importance of their results to their application.

3. Based on the results and the problem dimensions, the authors should discuss the possibilities of incorporating an unsupervised learning method into the problem.

"Minor editing of English language is required"

Reviewer 3 Report

The paper addresses an important topic of the automatic detection of anomalies in system logs. The authors propose a new method based on the BERT neural language model. The paper read well, and it looks technically solid.

My main concerns refer to the presentation of the results:

- There is no comparison of the results achieved with other studies on BGL or HDFS.

 - Are you sure the results in Figures 5 and 6 are statistically significant?

- The message conveyed by Fig. 2 is unclear.

- There is no need to quote BERT [13] several times; it is enough to do it when it is mentioned for the first time.

- Fig. 4 is careless: lines not aligned, dashed lines overlapping, mixed lower- and uppercase

- Table 5: wouldn't it be better visible in the form of a plot?

- "F1 − score" in Formula (8) suggests that it is a subtraction between F1 and score. Use "F1-score" instead.

- Missing articles and other language issues, e.g.,  "How is the generalization performance of CLDTLog?" => "What is the generalization performance of CLDTLog?", "data collection and Processing" => "data collection and processing". Please proofread the whole text.

- contrast learning => I'd suggest: contrastive learning

- L. 251 and the following: there should be a space between numbers and text, e.g., (1)Improving the robust... => (1) Improving the robust...

Round 2

Reviewer 3 Report

Thank you for addressing my comments. As for comparing with other studies, I meant referring to the results reported by others, I mean, e.g., studies like:

Wang, J.; Tang, Y.; He, S.; Zhao, C.; Sharma, P.; Alfarraj, O.; Tolba, A. LogEvent2vec: LogEvent-to-Vector Based Anomaly Detection for Large-Scale Logs in Internet of Things. Sensors 2020, 20, 2451.

Piotr Ryciak, Katarzyna Wasielewska, Artur Janicki. Anomaly Detection in Log Files Using Selected Natural Language Processing Methods. Applied Sciences. 2022; 12 (10):5089.

In my opinion, it is still missing, but it can be quickly added.
Anyway, the paper is much better and clearer now, and I think it can be published after the above minor corrections.